# Using Transferable Eucalypt Microsatellite Markers to Identify QTL for Resistance to *Ceratocystis* Wilt Disease in *Eucalyptus pellita* F. Muel. (Myrtales, Myrtaceae)

Heru Indrayadi [1,2,*], Morag Glen [1], Yusup Randy Kurniawan [2], Jeremy Todd Brawner [3], Bambang Herdyantara [2], Chris Beadle [1,4], Budi Tjahjono [2] and Caroline Mohammed [1]

1    Tasmanian Institute of Agriculture, University of Tasmania, Hobart, TAS 7001, Australia; morag.glen@utas.edu.au (M.G.); chris.beadle@csiro.au (C.B.); caroline.mohammed@utas.edu.au (C.M.)
2    Corporate R&D PT Arara Abadi—Sinarmas Forestry, Siak Regency, Perawang 28772, Riau, Indonesia; yusup.kurniawan@sinarmasforestry.com (Y.R.K.); bambang.herdyantara@sinarmasforestry.com (B.H.); budi.tjahyono@sinarmasforestry.com (B.T.)
3    Department of Plant Pathology, University of Florida, Gainesville, FL 32611, USA; jeremybrawner@gmail.com
4    CSIRO Ecosystem Sciences, Private Bag 12, Hobart, TAS 7001, Australia
*    Correspondence: heru.indrayadi@utas.edu.au

**Abstract:** The deployment of *Eucalyptus pellita* trees that are resistant to *Ceratocystis manginecans* is essential for the commercial plantations and therefore the sustainability of forest industries in Southeast Asia that utilize this resource. Current screening procedures are time-consuming and expensive but could be expedited with the aid of marker-assisted selection and breeding. The identification of genotypes with resistance to the disease may be facilitated if microsatellite markers developed in other *Eucalyptus* species are transferable and can be linked to quantitative trait loci (QTL) for disease resistance. This possibility was tested in 111 full-sib progenies and their parents by genotyping with 49 microsatellite markers developed in other *Eucalyptus* species. Disease development was assessed after stem inoculation with *C. manginecans* isolate Am60C. The disease index (DI) varied from 0 to 20% of stem length. There was a continuous distribution of resistant and susceptible seedlings with 60% in the resistant category. Of the 30 acceptable markers, 17 (56%) defined two linkage groups (LG). In each LG, one QTL with a significant logarithm of odds (LODs > 13) was identified. The transferability of microsatellite markers developed in other *Eucalyptus* species facilitated the rapid identification of LGs and QTLs in *E. pellita*. To further refine the linkage map, the testing of more microsatellite markers and a larger population of progenies are required.

**Keywords:** microsatellite transferability; *Ceratocystis manginecans*; *Ceratocystis fimbriata*

## 1. Introduction

Genetic markers allow for the early selection of characteristics in plant breeding programs and the eucalypt research community has developed more than 700 microsatellite markers for *Eucalyptus* species [1]. Developing genetic markers for resistance to wilt and canker disease caused by *Ceratocystis manginecans M. van Wyk, Al-Adawi & M.J. Wingfield* (Microascales, Ceratocystidaceae) will allow the selection of trees that will not be impacted by the disease in commercial plantations of *Eucalyptus pellita* [2–4]. In this study, the mapping of quantitative trait loci (QTL) was performed to identify chromosome regions that are associated with the resistance phenotype within a control pollinated family of *E. pellita* that were inoculated with *C. manginecans*. Microsatellite markers have been widely used for developing genetic markers in eucalyptus populations; however, few genomic studies have targeted *E. pellita* [5,6]. An advantage of microsatellite markers is their proven transferability among *Eucalyptus* spp. Within the Myrtaceae, markers have been transferred between *Eucalyptus grandis* W. Hill and *Eucalyptus urophylla* (L.A.S.Johnson ex

G.J.Leach) Brooker [7,8] as well as *E. grandis* and *Acca sellowiana* (O.Berg) Burret (Myrtales, Myrtaceae) [8–10]. Such transferability allows the same microsatellite markers to be used to construct linkage maps for many *Eucalyptus* species [8,11].

Genotypes of *E. pellita* vary in their levels of resistance to rust and leaf blight diseases as well as wilt and canker disease caused by *Ceratocystis* spp. [12]. Screening for resistance to *Ceratocystis fimbriata* Ellis & Halst in a clonal progeny of a *Eucalyptus* hybrid has resulted in reduced disease incidence [13,14]. Controlled pollination has been used to create full-sib families that have been used to identify superior trees and to evaluate the genetic control of disease resistance [15–17]. Narrow-sense heritability estimates of around 50% from those studies indicate that additive effects are significant in the disease tolerance of eucalypts therefore justifying a QTL approach [17,18]. The best clones for a specific trait can be selected efficiently from eucalypt families [19] as the progenies express a wide range of phenotypes and therefore form a highly informative population for genetic analysis [20,21]. For example, hybrid eucalypt clones of varying susceptibility to *C. fimbriata* were found in selection programs using full-sib progenies of *Eucalyptus* spp. [13,16].

High levels of phenotypic variability among resistant and susceptible genotypes have facilitated the identification of QTL in many species [3,15,22–24]. Quantitative trait locus analysis requires populations or progenies of a species which express phenotypic differences in a specific phenotypic trait [25]. For QTL mapping, the phenotypic scoring system for a particular trait must be applied to large numbers of individuals to obtain sufficient power to identify QTL that have a statistically significant effect on phenotype [22,23,26]. Using similar methods, five QTLs associated with resistance to *Ceratocystis* disease were discovered using 127 progenies of a *Eucalyptus* hybrid family [15].

Controlling *Ceratocystis* canker and wilt disease is challenging because it is soil-borne and can also spread in numerous ways [27,28]. *Ceratocystis* wilt and canker disease is already an important disease of planted *Eucalyptus* spp. in Brazil [29], China [30] and Vietnam [31], and has been identified in *Eucalyptus* spp. in Thailand and Indonesia [32]. In Indonesia and Malaysia, *E. pellita* is now the forest plantation species of choice [33,34], replacing the previous *Acacia mangium* resource that became non-commercial following attack by root rot disease caused by *Ganoderma philippii* (Bres. & Henn. ex Sacc.) Bres (Polyporales, Ganodermataceae) [35] and wilt and canker disease caused by *C. manginecans* [36–38]. Thus, *E. pellita* is being planted in areas with a high *Ceratocystis* inoculum load and developing populations that are resistant to *C. manginecans* is essential to the long-term viability of this new resource.

In this study, 50 microsatellite markers developed in *E. grandis*, *E. urophylla*, and *E. grandis * E. urophylla* were used to construct linkage maps and identify QTL positions in progenies from *E. pellita* parents. Resistance phenotypes were scored following inoculation with *C. manginecans*. Two hypotheses were tested: (1) that the progenies of a cross between resistant and susceptible *E. pellita* clones, SMAA7700EP * SMAA4990EP, demonstrate a wide range of resistance; (2) that the 50 microsatellite markers can be used to create a linkage map and locate a QTL conferring resistance or tolerance to *C. manginecans*.

## 2. Materials and Methods

### 2.1. Plant Materials

One hundred and eleven seedlings from control-pollinated seed derived from two *E. pellita* parents were vegetatively propagated using tissue culture to produce plantlets (produced with the same technique) that were screened for disease tolerance. The specific cross was between *E. pellita* clones SMAA7700EP and SMAA4990EP. The female parent, SMAA7700EP, was a tree selected from progeny trial EP14A in Riau Sumatra, established with seeds from Kiriwo, Papua New Guinea. SMAA4990EP was the male parent, selected from progeny trial EP03E, established with seed from Goe, Papua New Guinea. Seeds were obtained from the CSIRO Australian Tree Seed Centre in 1990 and 1998, respectively. Each rooted plantlet was transplanted into a 75 cm$^3$ tube containing a 2:1 mixture of cocopeat and rice husk and placed in an outdoor nursery for 90 days. The fertilizer Agrimore$^{\textregistered}$

(29-10-10 N:P:K) was applied twice a week in granular form at a concentration of 3 g/L. The plants were then moved into a shade house (70% light) for two weeks before inoculation. Between 4–6 ramets of each cloned progeny and the parents were inoculated and lesion length was measured to assess damage following inoculation with the pathogen.

## 2.2. Fungal Inoculation and Phenotypic Evaluation

A single isolate of *C. manginecans*, Am0160C, was used for all inoculations. It was originally isolated from *A. mangium* and has been used for the pathogenicity screening of acacias and *E. pellita* in previous tests. The isolate was cultured in 9 mm diameter Petri dishes containing Potato Sucrose Agar prepared from 200 g of boiled potato to produce 1 L of broth which was then mixed with 20 g of sucrose and 20 g of agar and autoclaved for 20 min at 121 °C. Agar plugs measuring 3 mm in diameter taken from the growing margin of 7-day-old cultures were used as the inoculum; a 3 mm cork borer was used to prepare the plug and to wound the plants 3 cm from the stem's base. After inoculation, parafilm® (Bemis, Sheboygan Falls, WI, USA) was used to seal the wound. Sixty days after inoculation, the bark was carefully removed above and below the inoculation point, and the inner lesion length was measured [39]. The disease index (DI) was estimated as the ratio of the internal lesion (IL) length to the total stem length, where DI = 100 × IL/plant height. The significance ($p < 0.05$) of differences in DI among the clones was evaluated via an analysis of variance in R.

## 2.3. Microsatellite Genotyping

Microsatellite loci that were developed for *Eucalyptus* spp. [7–9,11,40,41] were used (Table 1). DNA was extracted from progenies and parents using the CTAB method, as modified by Grattapaglia and Sederoff [42]. The concentration of DNA was measured in a NanoDrop 2000 spectrophotometer (Thermo Fisher Scientific, Inc., Waltham, MA, USA); each sample was then diluted to 5 ng/μL. Each PCR reaction had a total volume of 10 μL, comprising 1× Platinum PCR buffer (Invitrogen, Waltham, MA, USA), 1.5 mM MgCl2, 0.2 mM dNTPs, 0.5 μM of each primer, 0.25 U-Platinum Taq polymerase (Invitrogen, USA) and approximately 20 ng of the DNA sample. PCR amplification was conducted on either a GeneAmp 9700 (Applied Biosystem, Waltham, MA, USA) or Veriti (Applied Biosystem, Waltham, MA, USA) Thermal Cycler using the following program, modified from Brondani [9] (Suharyanto pers. comm.): initial denaturation at 94 °C for 2 min followed by 35 cycles of denaturation (94 °C for 60 s), annealing (56 °C for 60 s), extension (72 °C for 60 s) and finally post-extension at 72 °C for 7 min.

**Table 1.** Forty-nine microsatellite markers used for genotyping the *E. pellita* population.

| Marker (Prefix) | Marker Number | Original Species Target | Reference |
|---|---|---|---|
| EUCeSSR | 010, 070, 130, 131, 165, 227 | *E. urophylla*, *E. tereticornis* Sm *E. grandis*, *E. saligna* Sm, *E. camaldulensis* | [41] |
| EMBRA | 925, 1307, 1364, 1811, 2014 | Brooker & Hopper, *E. urophylla*, *E. globulus* Labill, *E. dunnii* Maiden | [11,40] |
| EMBRA | 5, 6, 20 | *E. grandis*, *E. urophylla* | [9] |
| EMBRA | 21, 23, 28, 29, 31, 36, 40, 41, 42, 47, 56 114, 155, 156, 157, 158, 159, 169, 172, | *E. grandis*, *E. urophylla* | [8] |
| EMBRA | 184, 191, 195, 207, 239, 269, 277, 279, 301, 302, 350, 351, 368, 369, 372, 378 | *E. grandis*, *E. urophylla* | [7] |

PCR products were separated and visualized using polyacrylamide gel electrophoresis (PAGE) with silver staining. A 100-well comb was used on each PAGE plate for sample loading to enable the simultaneous analysis of samples. A DNA size standard (10-bp ladder, Thermo Scientific, Inc., Waltham, MA, USA) was employed on the left- and right-hand sides of the plate. Allelic calling at all loci was conducted using Quantity One software 4.6.2 (Bio-Rad. Inc., Hercules, CA, USA) [43].

### 2.4. Genetic Linkage Map Analyses

The genetic linkage map was analyzed using R/qtl following the methods of Broman [44]. Any markers which had an error in genotyping of 111 progenies and two parents were excluded. A Chi-squared analysis ($p < 0.05$) was used to test the markers for the Mendelian segregation of all progenies involved for F2 in a ratio of 1:2:1 and to make an association [44]. Pairwise marker linkages were arranged with a minimum logarithm of odds (LOD) score of 3.5 and a recombination frequency of 0.35. Marker distances were calculated in accordance to the mapping function of Haldane [45] and Kosambi [46].

### 2.5. QTL Analysis

The R/qtl package was used to analyze QTL mapping results. The phenotype (DI) of the 111 progenies and two parents was used to analyze the importance of QTL and the linkage map was used to evaluate the position of QTL. The LOD thresholds for each QTL were computed using permutation analysis with 100 repetitions at a 5% probability ($p < 0.05$). The peak in each linkage group or QTL position was then identified and summarized. Finally, using MapChart 2.32 software [47], a graph of the QTL position in each linkage group was drawn.

## 3. Results

### 3.1. Phenotypic Evaluation

There were significant differences ($p < 0.0001$) among the clonally replicated progeny and their parents. The mean DI of individual progeny ranged from 0.85 to 20.65%; the mean DIs for all clones was $5.18 \pm 0.18$%. The mean DIs of each parent were $3.01 \pm 0.40$ for SMAA7700EP and $7.42 \pm 0.32$ for SMAA4990EP. The DIs were significantly different among the clones ($p < 0.05$). The progenies were divided into five continuous groups (Figure 1) using a grouping strategy based on the Scott–Knott analysis of the DI [48]. Sixty-seven (~60%) of the progenies were in the resistant group (Group 1) and fifteen were in the susceptible and highly susceptible groups (Groups 3–5) (Figure 1). There was a positive correlation (r = 0.96) between the DI and the inner lesion length, IL (Figure 2).

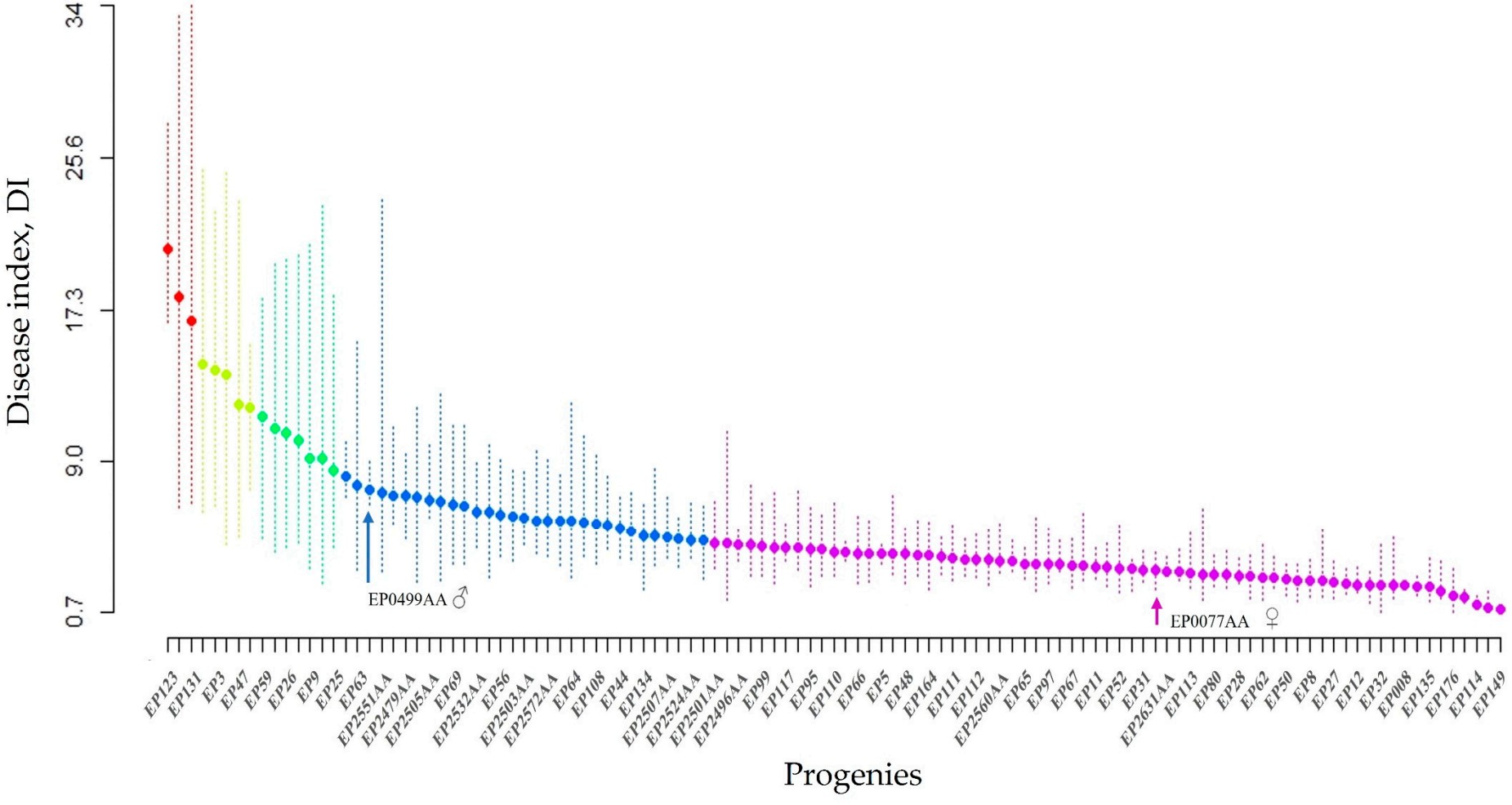

**Figure 1.** Disease index (DI) of 111 progenies and their parents; SMAA7700EP as the female parent was assigned to the first group and SMAA4990EP as the male parent was assigned to the second group. The groups were classified as resistant (violet—group 1), intermediate (blue—group 2), susceptible (yellow and green—group 3 and 4), and highly susceptible (red—group 5).

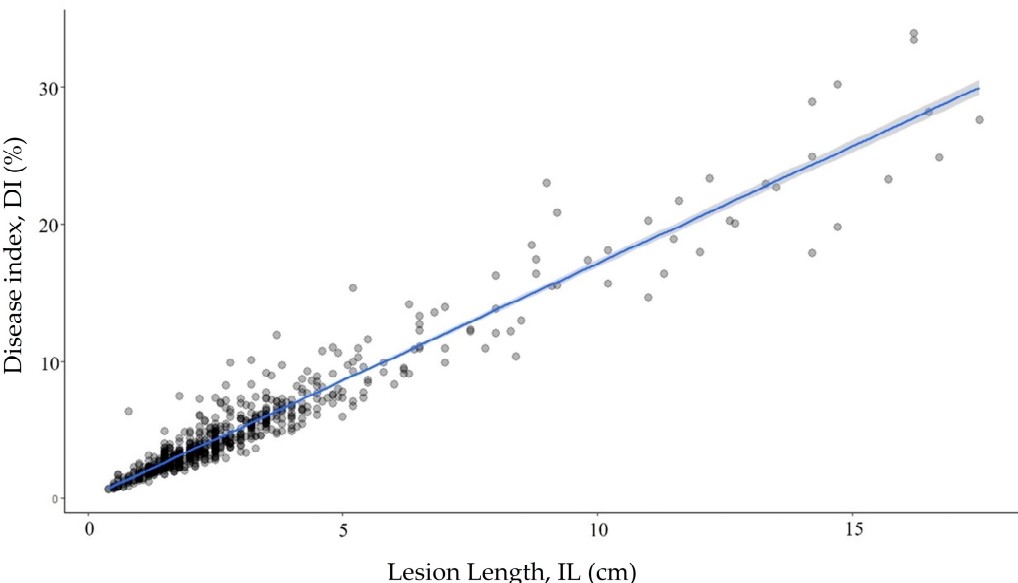

**Figure 2.** Linear correlation of the DI with the inner lesion length demonstrating the similarity in assessments.

### 3.2. Genotyping and Linkage Map Analysis

Out of the 49 microsatellite marker primer pairs tested, 19 markers (39%) were excluded via R/qtl analysis. The majority of these (15 markers) produced non-specific amplificons in addition to the target microsatellite, while 2 were polyallelic and the remaining 2 had null alleles. This left thirty markers or 61% of those tested for constructing the linkage map. These comprised 25 di-nucleotide and 5 tetra- or hexa-nucleotide microsatellites. Of these, seventeen (56%) (Table 2) formed a linkage map with two linkage groups (LG) (Figure 3). Three of the five tetra- and hexa-nucleotide (60%) markers were included in the LGs. The total distance covered by the map was 509.9 centimorgans (cM); for LG1 it was 399.3 (cM) with an average distance of 36.3 cM and for LG2 it was 110.6 cM with an average distance of 18.6 cM.

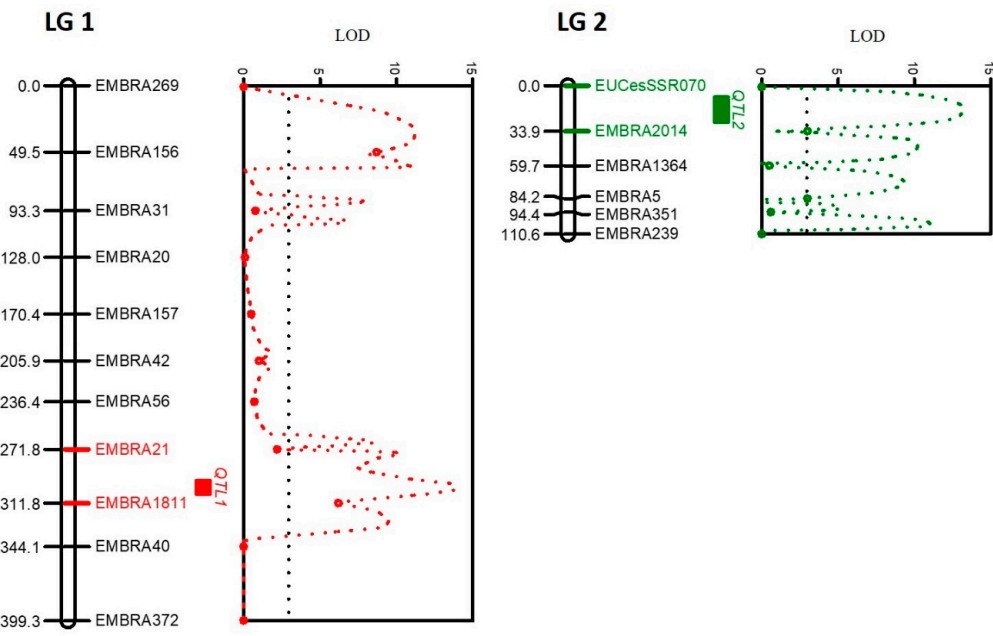

**Figure 3.** The corresponding QTL positions in the two different linkage groups for *Ceratocystis* disease

resistance in the mapping of *E. pellita* (SMAA7700EP) * *E. pellita* (SMAA4990EP). Red (LG1) and green (LG2) bars indicate the markers that encompass the QTL positions with the highest LOD shown on the graphs.

**Table 2.** The seventeen markers used to construct the two linkage groups.

| Primer ID | | Primer Sequences 5′ to 3′ | LG * | SSR Type | Allele Size Range |
|---|---|---|---|---|---|
| EMBRA5 | F | ATGCTGGTCCAACTAAGATT | 5 | Di | 117–130 |
| | R | TGAGCCTAAAAGCCCAAC | | | |
| EMBRA20 | F | GTGAGTGGGTATCCATCG | 6 | Di | 136–154 |
| | R | GCTGGAACTGGTCTTGAG | | | |
| EMBRA21 | F | ACAAGGGAAACTTGATCG | 10 | Di | 148–150 |
| | R | GGAACCGAACATAGCAAG | | | |
| EMBRA31 | F | AATTGCCCGAGTCAAAATAC | 6 | Di | 129–145 |
| | F | GGAACAATGTGGTTTGGG | | | |
| EMBRA40 | R | AAAGTATCTTCACGCTTCAT | 10 | Di | 120–146 |
| | F | TCCCAATCATGATCTTCAG | | | |
| EMBRA42 | R | GAGTAAAAATTGGTTTTGAGTG | 7 | Di | 115–126 |
| | F | CCCTCTTTTCATTTTGTCTT | | | |
| EMBRA56 | R | TCATTGACATGCTGACTGT | 1 | Di | 127–143 |
| | F | ACTAACAGTTGAAAAGGTAAAGC | | | |
| EMBRA156 | F | GTCAGATTGGATCTATGC | 4 | Di | 115–117 |
| | R | GAACAAGTAGATCCTCGTA | | | |
| EMBRA157 | F | TGCCAGAATGTATCGTCC | 8 | Di | 128–152 |
| | R | TCTGGCTTCTTTCTTGTTG | | | |
| EMBRA239 | F | AAGAGAGAGTGATTGGCGAG | 3 | Di | 174–193 |
| | R | CTGTGACACTAGGCATGTTG | | | |
| EMBRA269 | F | TCAACTGCAATCCTTACC | 11 | Di | 194–206 |
| | F | CCTGCAGTGTCAGTGTGT | | | |
| EMBRA351 | R | CTAGGTGAGGGAAATGAAA | - | Di | 108–110 |
| | F | CCAGACAACAAGAAGAAAGT | | | |
| EMBRA372 | R | ACTTTGGATTACCCGCTATA | - | Di | 138–148 |
| | F | CATTCTGATTCCACCGTATA | | | |
| EMBRA1364 | R | CGTTTTCGCTCCTCTCTCTC | 8 | Tetra | 580–595 |
| | F | TGTAGAGATCGGGGTCCTTG | | | |
| EMBRA1811 | F | GTCGAGTTGAGTTCGCTTCC | 9 | Hexa | 272–300 |
| | R | AGTGAATCGGGAGAGGAGGT | | | |
| EMBRA2014 | F | CACCGACTTCCTCTTCTTCG | 9 | Tetra | 117–129 |
| | R | CCCCATCCCTTCTCTCTCTC | | | |
| EUCeSSR070 | F | AAACTAAGCTGGGAGGAA | - | Di | 182–189 |

* = LG, linkage group based on the consensus map of *E. grandis* [7–9].

*3.3. QTL Mapping*

The QTL position in LG1 was between EMBRA1811 and EMBRA21 with a significant LOD of 13.9 (threshold ≥ 3); it was located at locus 298 at 13.8 cM from EMBRA1811 (Figure 3). The QTL position in LG2 was between EMBRA2014 and EUCeSSR070 with LOD = 13.2, also significant; it was detected at locus 18 at 15.9 cM from the marker

EMBRA2014 (Figure 3). An interaction between EMBRA1811 and the DIs in LG1 showed that B allele was linked to greater disease susceptibility (Figure 4A). An interaction between EMBRA2014 and the DIs in LG2 showed that allele A was a marker for greater disease tolerance (Figure 4B).

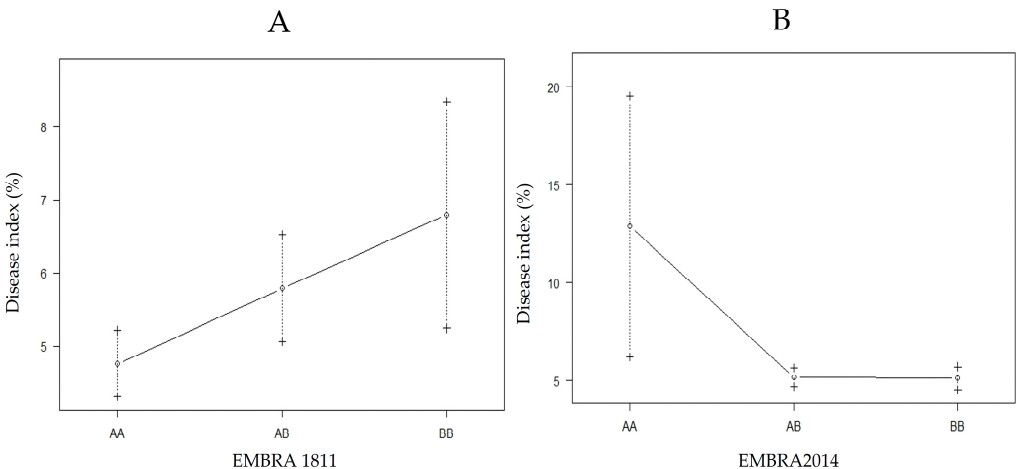

**Figure 4.** Interaction between the nearest marker (genotypes) to the QTL position with the DIs in LG1 (**A**) and LG2 (**B**); +/−: standard error.

## 4. Discussion

The first hypothesis was confirmed; a significant variation in the DI (0.85 to 20.65%) was found among clones. More than half of the progenies were classified as resistant to *C. manginecans* infection, confirming the choice of *Eucalyptus pellita* as a potential source of resistance to *C. manginecans*. This species is also a source of resistance to *C. fimbriata* [12], as indicated by at least half the clones tested being resistant to this disease. This is in contrast to the case of *Acacia mangium*, where resistant clones are very rare [38]. Variability in disease resistance to *Ceratocystis* wilt disease has been found previously in progenies of *E. grandis * E. urophylla* [13,16] and *Mangifera indica* L (Sapindales, Anacardiaceae) [49]. This study indicates that this is also the case for *E. pellita* and resistance to *C. manginecans;* there is, therefore, the potential for selecting resistant clones via the early screening offspring obtained through controlled pollination, including at the seedling stage.

The strong correlation between the DI and inner lesion length, IL ($r = 0.96$), indicates that either variable may be used for evaluating disease severity, providing plants are all of similar height. To accurately assess disease infection using artificial inoculation in a greenhouse, the methods that have been adopted must achieve consistent and comparable results. For the phenotypic measurement of *Ceratocystis* diseases in *Eucalyptus* spp., either xylem discoloration or the IL after artificial inoculation was first used [12,16,50]. More recently, the DI has been adopted to represent disease severity [15,51,52]. All these methods have proven reliable for measuring pathogenicity using vegetatively propagated plants in a nursery. When evaluating symptoms using either the inner lesion or the DI, factors such as plant height, species, and the sources of the plant (seedling or grafting) may be taken into consideration. Evaluating potted plants that have sufficient vascular tissue to measure lesion development in the nursery may be used to increase the efficiency of in-field disease resistance screening.

The second hypothesis was also confirmed; of the 49 microsatellite markers tested in this study that had been developed in *E. grandis*, *E. urophylla*, and *E. grandis * E. urophylla*, 61% were readily transferred to *E. pellita*, and of these, 17 could be used to construct a linkage map. This met the expectation that eucalypt species have a microsatellite transferability rate of between 40 and 90% [53], and showed that primer pairs developed to date in these other eucalypt species [7,54] have good transferability to *E. pellita*, despite the apparent lack of synteny between *E. pellita* and other eucalypt species. A higher transfer

rate may have been achieved if PCR conditions had been optimized to avoid non-specific amplification though the testing of a greater number of markers may be a more efficient strategy. Microsatellite loci with di- and tri-nucleotide repeats have been used more often for genotyping studies because they usually have higher variability and variation among individuals [1], although they frequently show multiple bands or stutter peaks [55]. Although less polymorphic, tetra-, penta-, and hexa-nucleotide repeats are more reliable for constructing linkage maps because these markers have a greater precision of allelic calling [1,40]. The construction of an improved linkage map covering all chromosomes in *E. pellita* will require a greater number of markers. In the absence of *E. pellita* genomic resources, marker transfer from other *Eucalyptus* species is more efficient than is developing specific markers for *E. pellita*. High transferability also facilitates the integration of genetic linkage maps among *Eucalyptus* species and may provide an efficient approach to identifying QTL [56]. Of particular interest are the flanking markers that may be used for other traits such as disease tolerance, wood properties and drought tolerance [15,57–59]. The likelihood of obtaining additional transferrable markers for *E. pellita* is high due to the availability of more than 700 microsatellite markers for the genus *Eucalyptus* [1].

The low-density genetic map formed by the seventeen markers in this study had an average distance between markers that was >18 cM. Marker numbers determine the density of the map [60,61] and low-density maps are associated with inconsistency in the determination and recognition of QTL [58,62]. In the consensus map of *E. grandis* with an average distance between markers of 8.4 cM, 10 of the 17 markers that contributed to the two linkage maps of *E. pellita* in this study have also been found in separate linkage groups, in *E. grandis* [7,11]. Four of the seventeen markers, EMBRA269, 1364, 1811, and 2014, have been used in the construction of a consensus map [11,54,58]. Three, EMBRA351, EMBRA372 and EUCeSSR071, have not previously been incorporated into linkage maps for other eucalypt species. Constructing a robust genetic linkage map for *E. pellita* is required to enhance the effectiveness, efficiency, and precision of QTL analysis as well as the capacity to identify candidate specific genes associated with selected traits [63–65]. Breeders can use precise markers that are strongly linked to the desired traits in marker-assisted selection (MAS) programs; this will expedite the development of germplasm by reducing the number of clones for phenotypic testing, removing unreliable environmental effects, and selecting for specific traits at the seedling stage [22,61]. Thus, the development of a higher-density linkage map will enhance *E. pellita* breeding programs.

Two QTL positions were found in two separate LGs using 111 progenies from two parents and 30 pairs of microsatellite primer markers. This finding aligned with the prediction made based on the population size that a QTL should be found in a population of 100 or more progenies from a controlled cross [66]; however, the number of QTL in each LG is not dependent on the number of microsatellite markers involved [22]. In an examination of *Ceratocystis* disease resistance in a eucalypt hybrid (*E. dunnii* * *E. grandis*) * (*E. urophylla* * *E. globulus*), 127 plants of F2 progenies and 114 marker pairs led to the discovery of five QTL in different positions across the 11 linkage groups [7,15]. Even though the flanking markers differed, the QTL position for *Ceratocystis* disease resistance was in the same linkage group, LG1, in both studies. No QTL were identified in LG2 of the hybrid. The QTL within LG2 of this *E. pellita* family is primarily linked to disease susceptibility with two of the five QTL found in the hybrid also linked to susceptibility [15]. In an unpublished study, one QTL in *E. pellita* that was also in LG1, was linked to resistance to bacterial wilt caused by *Ralstonia psedosolanacearum* (Bayo A. Siregar, pers. comm.). Both studies used the same markers and progenies and EMBRA21 was a common flanking marker. As noted above, *E. pellita* has been shown to be resistant to three diseases, including *Ceratocystis* wilt disease [12], and potentially to bacterial wilt disease (Bayo A. Siregar, pers.comm.).

## 5. Conclusions

This study has shown that a wide range of resistance to *C. manginecans* could be found within a full-sib family of *E. pellita*. Secondly, the transferability of existing microsatellite

markers in eucalypts enabled the identification of QTLs in two linkage groups associated with resistance or susceptibility. These QTLs provide a relative position for a gene or gene clusters that may be used to guide selection for disease resistance [67,68]. It is more informative to map traits and QTL in the progeny of families from controlled crosses between disparate populations because of the increase in genetic variation compared to that in the parental populations. High marker density and association mapping in many segregating families will be required for more precise QTL location [69,70]; this will improve the resolution of the candidate genes. Further research is required to combine the findings of this study with data from a variety of omics platforms [71], such as transcriptomic analysis which has shown that cell wall degrading enzymes (CWDEs) are the major genes responsible for *Ceratocystis fimbriata*'s pathogenicity in eucalypts [72]. The identification of genes linked to host resistance or susceptibility will enhance our understanding of host–pathogen interactions and provide reliable breeding markers. The availability of a high-density linkage map will also facilitate selection for other desirable traits such as fibre quality.

**Author Contributions:** Conceptualization, methodology, formal analysis, and writing—original draft, H.I., M.G. and C.B.; methodology and formal analysis, H.I. and Y.R.K.; writing—review and editing, H.I., C.B., J.T.B., M.G. and C.M.; funding acquisition, investigation and license, C.M., M.G., B.T. and B.H. All authors have read and agreed to the published version of the manuscript.

**Funding:** Sinarmas Forestry (SMF) provided materials and technical assistance. The Australian Centre for International Agricultural Research provided financial support through FST/2014/068 and a John Allwright Fellowship for Heru Indrayadi.

**Institutional Review Board Statement:** This study does not examine any human or animal subjects.

**Data Availability Statement:** The data presented in this study are available on request from the corresponding author. The data are not publicly available due to company policy.

**Acknowledgments:** We express our gratitude to the Australian Centre for International Agricultural Research and the management of R&D Sinarmas Forestry for supporting this study. Heru Indrayadi is the recipient of a John Allwright Fellowship. Thanks also go to Suharyanto for discussions and to R&D Corporate, Sinarmas Forestry for the microsatellite data.

**Conflicts of Interest:** The authors declare there are no conflict of interest.

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
