# Peer review of "Using Transferable Eucalypt Microsatellite Markers to Identify QTL for Resistance to Ceratocystis Wilt Disease in Eucalyptus pellita F. Muel. (Myrtales, Myrtaceae)"

_forests, doi:10.3390/f14091703_

Round 1

Reviewer 1 Report (Previous Reviewer 1)

They have given serious consideration to my comments and I largely accept their argument. 

na

Author Response

Reviewer 1 did not add any new comments

Reviewer 2 Report (New Reviewer)

Journal: Forests (ISSN 1999-4907)

Manuscript ID forests-2535295

Type of the Paper: Communication

Manuscript Title: "Using transferable eucalypt microsatellite markers to identify QTL for resistance to Ceratocystis wilt disease in Eucalyptus pellita"

Reviewer comments:

The current work reveled using transferable eucalypt microsatellite markers to identify QTL for resistance to Ceratocystis wilt disease in Eucalyptus pellita. After carefully reviewing the present work, I feel that this work is suitable for publication, properly executed. Therefore, only I have some comments which might be helpful in improving this work. On the other hand, this is a well-designed, well-referenced and well-written article. However, I have some concerns which authors should take in consideration. Besides my suggestion, a thorough proof-check (punctuation, spelling/typing) and most importantly English language is recommended.

  • Abbreviations should not be written in the title of the research unless the abbreviation is known internationally among the scientific community, as this can be overlooked in this case. The first appearance of abbreviations needs to be marked with complete definitions. Pls pay attention to this comment in the similar cases in the entire manuscript. This paper utilizes lots of acronym names. Why not establish a listing table for all the abbreviation used along with their full names.
  • Please, I would like to know, is this technique suitable for detecting disease infections of some very important forest plants, for example, the Dieback disease that affects juniper plants in various regions of the Kingdom of Saudi Arabia? If yes, this will be important technique in this concern.
  • The abstract needs to be further simplified and highlight innovation. After reading the abstract, readers should have a general understanding of the full text and be clear about motivation and breakthrough. Please pay attention to this comment.
  • L 21: “we” Throughout the manuscript, avoid using any personal pronouns.
  • First letter of key words should be capital.
  • L 35: delete “quantitative trait loci (  )”.
  • For Scientific names of all organisms as in lines 36, 37, it should add Orders and Families when they are mentioned for the first time
  • Lines 44 and 45: it should mention the scientific name in full Genera names when they are mentioned for the first time. Do these for all organisms in all the manuscript
  • L 53-55: “Narrow-sense heritability estimates of around 50% from those studies indicate that additive effects are significant in disease tolerance of eucalypts therefore justifying a QTL approach”, Pls add relevant reference
  • L 64: “(Ammitzboll et al., 2018)” delete and rewrite according to the Journal instruction, the rearrange the following numbers [24, 25, ……………..] in the following parts of the manuscript.
  • L 86: “C. manginecans” have to be italic.
  • L 90-93: pls correct “One hundred and eleven seedlings from control-pollinated (CP) seed derived from two E. pellita parents were vegetatively propagated using tissue culture to produce plantlets that were screened for disease tolerance. Both parents used to produce the CP seed were captured and propagated using tissue culture” to “One hundred and eleven seedlings from control-pollinated (CP) seed derived from two E. pellita parents were vegetatively propagated using tissue culture to produce plantlets (produced with the same technique) that were screened for disease tolerance.
  • L 110: “Potato Sucrose Agar (PSA)” this term mentioned one time, so, no need to abbreviate it
  • L 117: “Olivera et al. (2016)” delete and rewrite according to the Journal instruction, the rearrange the following numbers [……, ……………..] in the following parts of the manuscript. Line 117: Olivera et al. is missed in references list. Add it as number without the year. The same comment for lines 122 and 123.
  • Lines 131-139: these PCR conditions need to add reference
  • Line 143-144: Broman [40] instead of Broman, K. W., H. Wu, S. Sen and G. A. Churchill [40]
  • Line 145: Add the reference of Chi-squared analysis
  • He et al. 2012 and other others in the same Table 1” pls write the number only according to the Journal instructions.
  • Follow the author guidelines when you mentioned citation in all MS, most of them are incorrect as in line 146
  • L 149-150: “Haldane, J. B. S. [41] and Kosambi, D. D. [42] mapping function.” Remove, J. B. S. and, D. D.
  • L 162: delete “disease index (  )
  • L 168 “(Jelihovschi, Faria & Allaman 2014)” delete and rewrite according to the Journal instruction, the rearrange the following numbers [……, ……………..] in the following parts of the manuscript.
  • Hypothesis of the work should be clear at the end of the introduction part.
  • Some paras are in the introduction part, try to lower them to few paras
  • Resolution of Figure (1) is poor. Kindly improve.
  • Figure (2), same.
  • L 243: “disease index” write the abbreviation only.
  • In the discussion section. Please briefly describe the results in the first paragraph of discussion, the discuss this result and finally support this discussion by previous studies, pls follow this in the entire discussion section.
  • Discussion needs to be improved. Please discuss the possible mechanism behind your results. Avoid comparing with others work in each paragraph.
  • For the discussion part, please end with a summary of conclusions regarding the significance of the work and suggestions for future studies.
  • L 350: Author contribution, pls write this part according to the Journal instructions.
  • The conclusion is an important part of this manuscript, pls write a conclusion and add future perspectives at the end of it.
  • The scientific names have to be italic in the entire manuscript.
  • The references’ part is written in a manner contrary to the writing method of the MDPI Journals. The whole references section has to be rewrite according to the journal instructions. Pls RW.
  • All sections need to revised as linguistic editing
  • Overall, this article is still interesting, and it is recommended to accept it after major revision.

 Extensive editing of English language required

Author Response

Dear Editor

            We thank the reviewer for their time and care in providing feedback on this manuscript. Recommended changes have been implemented as detailed below.

Reviewer comments: 

The current work revealed using transferable eucalypt microsatellite markers to identify QTL for resistance to Ceratocystis wilt disease in Eucalyptus pellita. After carefully reviewing the present work, I feel that this work is suitable for publication, properly executed. Therefore, only I have some comments which might be helpful in improving this work. On the other hand, this is a well-designed, well-referenced and well-written article. However, I have some concerns which authors should take in consideration. Besides my suggestion, a thorough proof-check (punctuation, spelling/typing) and most importantly English language is recommended. 

  • Abbreviations should not be written in the title of the research unless the abbreviation is known internationally among the scientific community, as this can be overlooked in this case. The first appearance of abbreviations needs to be marked with complete definitions. Pls pay attention to this comment in the similar cases in the entire manuscript. This paper utilizes lots of acronym names. Why not establish a listing table for all the abbreviation used along with their full names. 

There are only 6 acronyms in this paper, three of which are in common use by geneticists. We have not prepared a table of abbreviations but can do so if required. The instructions to authors do not specify where such a table should be located.

  • Please, I would like to know, is this technique suitable for detecting disease infections of some very important forest plants, for example, the Dieback disease that affects juniper plants in various regions of the Kingdom of Saudi Arabia? If yes, this will be important technique in this concern. 

We are unable to answer this question as it is unclear which technique is referred to. Recent publications indicate that the juniper dieback in Saudi Arabia is linked to climatic changes.

  • The abstract needs to be further simplified and highlight innovation. After reading the abstract, readers should have a general understanding of the full text and be clear about motivation and breakthrough. Please pay attention to this comment. 

More detailed explanation has been provided in the abstract.

  • L 21: “we” Throughout the manuscript, avoid using any personal pronouns. 

All personal pronouns have been removed.

  • First letter of key words should be capital. 

Keywords have been capitalised.

  • L 35: delete “quantitative trait loci (  )”. 

Sentence has been modified to remove any reference to QTLs. Author guidelines (https://www.mdpi.com/journal/forests/instructions#preparation) state;

Acronyms/Abbreviations/Initialisms should be defined the first time they appear in each of three sections: the abstract; the main text; the first figure or table. When defined for the first time, the acronym/abbreviation/initialism should be added in parentheses after the written-out form.

  • For Scientific names of all organisms as in lines 36, 37, it should add Orders and Families when they are mentioned for the first time 

Added

  • Lines 44 and 45: it should mention the scientific name in full Genera names when they are mentioned for the first time. Do these for all organisms in all the manuscript 

Included

  • L 53-55: “Narrow-sense heritability estimates of around 50% from those studies indicate that additive effects are significant in disease tolerance of eucalypts therefore justifying a QTL approach”, Pls add relevant reference 

Added

  • L 64: “(Ammitzboll et al., 2018)” delete and rewrite according to the Journal instruction, the rearrange the following numbers [24, 25, ……………..] in the following parts of the manuscript. 

Corrected

  • L 86: “C. manginecans” have to be italic. 

Corrected

  • L 90-93: pls correct “One hundred and eleven seedlings from control-pollinated (CP) seed derived from two E. pellita parents were vegetatively propagated using tissue culture to produce plantlets that were screened for disease tolerance. Both parents used to produce the CP seed were captured and propagated using tissue culture” to “One hundred and eleven seedlings from control-pollinated (CP) seed derived from two E. pellita parents were vegetatively propagated using tissue culture to produce plantlets (produced with the same technique) that were screened for disease tolerance.” 

Change accepted

  • L 110: “Potato Sucrose Agar (PSA)” this term mentioned one time, so, no need to abbreviate it 

Abbreviation removed

  • L 117: “Olivera et al. (2016)” delete and rewrite according to the Journal instruction, the rearrange the following numbers [……, ……………..] in the following parts of the manuscript. Line 117: Olivera et al. is missed in references list. Add it as number without the year. The same comment for lines 122 and 123. 

All corrected

  • Lines 131-139: these PCR conditions need to add reference 

These are the conditions that were used in this study, no reference is required.

  • Line 143-144: Broman [40] instead of Broman, K. W., H. Wu, S. Sen and G. A. Churchill [40] 

Corrected

  • Line 145: Add the reference of Chi-squared analysis 

Added

  • “He et al. 2012 and other others in the same Table 1” pls write the number only according to the Journal instructions. 

Corrected

  • Follow the author guidelines when you mentioned citation in all MS, most of them are incorrect as in line 146 

Corrected

  • L 149-150: “Haldane, J. B. S. [41] and Kosambi, D. D. [42] mapping function.” Remove, J. B. S. and, D. D. 

Removed

  • L 162: delete “disease index (  )” 

Removed

  • L 168 “(Jelihovschi, Faria & Allaman 2014)” delete and rewrite according to the Journal instruction, the rearrange the following numbers [……, ……………..] in the following parts of the manuscript. 

All reference formatting has been corrected.

  • Hypothesis of the work should be clear at the end of the introduction part. 

Two hypotheses are stated clearly.

  • Some paras are in the introduction part, try to lower them to few paras 

The number of paragraphs has been reduced from 6 to 5.

  • Resolution of Figure (1) is poor. Kindly improve. 

The figure has been replaced, though size limitations make the clone codes difficult to read. A larger figure has been supplied as supplementary material.

  • Figure (2), same. 

The figure has been replaced.

  • L 243: “disease index” write the abbreviation only. 

Revised

  • In the discussion section. Please briefly describe the results in the first paragraph of discussion, the discuss this result and finally support this discussion by previous studies, pls follow this in the entire discussion section. 

We have not followed this format, rather we present each experiment and discuss its implications and relationship to previous studies as this gives a more logical flow and avoids repetition.

  • Discussion needs to be improved. Please discuss the possible mechanism behind your results. Avoid comparing with others work in each paragraph. 

The discussion has been revised, language and conceptual flow improved and superfluous material removed.

  • For the discussion part, please end with a summary of conclusions regarding the significance of the work and suggestions for future studies. 

This has been added.

  • L 350: Author contribution, pls write this part according to the Journal instructions. 

Done

  • The conclusion is an important part of this manuscript, pls write a conclusion and add future perspectives at the end of it. 

Done

  • The scientific names have to be italic in the entire manuscript.

Corrected

  •  The references’ part is written in a manner contrary to the writing method of the MDPI Journals. The whole references section has to be rewrite according to the journal instructions. Pls RW. 

Corrected

  • All sections need to revised as linguistic editing 

English language has been checked and corrected by a native English speaker.

  • Overall, this article is still interesting, and it is recommended to accept it after major revision. 

    Comments on the Quality of English Language 

     Extensive editing of English language required 

Language has been corrected where necessary.

Round 2

Reviewer 2 Report (New Reviewer)

After carefully revising the corrected version in the current manuscript, I have some concerns which authors should take in consideration as follows:

  • Keywords need to be reconsidered. The authors have to write words that are the focus of this research.
  • In the materials and methods section, PCR conditions need to add reference 
  • Hypothesis of the work should be clear in the introduction part (at the end). 
  • Please briefly describe the results in the first paragraph of discussion, the discuss this result and finally support this discussion by previous studies, pls follow this in the entire discussion section. 
  • Extensive editing of English language required 
  • Overall, this article is still interesting

Author Response

Dear Editor

            We appreciate the reviewer's effort and consideration in giving this document an objective assessment. The modifications that were advised have been made are listed below.

Reviewer comments: 

After carefully revising the corrected version in the current manuscript, I have some concerns which authors should take in consideration as follows:

  • Keywords need to be reconsidered. The authors have to write words that are the focus of this research.

There is no point adding keywords that are already in the title or abstract as these are already indexed

  • In the materials and methods section, PCR conditions need to add reference 

Added

  • Hypothesis of the work should be clear in the introduction part (at the end). 

Two hypotheses are explicitly stated at the end of the Introduction (lines 89-93)

Two hypotheses were tested: (1) that progeny of a cross between resistant and susceptible E. pellita clones, SMAA7700EP x SMAA4990EP, demonstrate a wide range of resistance; (2) the 50 microsatellite markers can be used to create a linkage map and locate a QTL conferring resistance or tolerance to C. manginecans”

  • Please briefly describe the results in the first paragraph of discussion, the discuss this result and finally support this discussion by previous studies, pls follow this in the entire discussion section. 

This format is not specified in the journal’s ‘Instructions to Authors’ which stipulates;

  • Discussion: Authors should discuss the results and how they can be interpreted in perspective of previous studies and of the working hypotheses. The findings and their implications should be discussed in the broadest context possible and limitations of the work highlighted. Future research directions may also be mentioned. This section may be combined with Results.

The Results have already been presented in the Results section. This suggestion would have us repeating them immediately afterwards at the start of the Discussion. It is more logical, in the Discussion, to mention the outcomes from each experiment and discuss them within the context of previous work.

Please read each first paragraph thoroughly as the results are briefly explained there. For instance, lines 173–174 in the result and 246 in the discussion; lines 180 in the result and 258 in the discussion; etc.

In addition, recent papers published in Forests follow a similar format to ours, For example Zitha et al., 2023  https://www.mdpi.com/1999-4907/14/8/1661 and Awing et al 2023 https://www.mdpi.com/1999-4907/14/8/1660

  • Extensive editing of English language required 

The English language has been carefully edited by a native English speaker, which this reviewer clearly is not.

  • Overall, this article is still interesting

This manuscript is a resubmission of an earlier submission. The following is a list of the peer review reports and author responses from that submission.

Round 1

Reviewer 1 Report

Indrahyadi et al

Using transferable eucalypt microsatellite markers to identify QTL for resistance to Ceratocystis wilt disease in Eucalyptus

The paper reports a study to detect QTL explaining genetic variation to Ceraocystis resistance in an E. pellita control pollinated family.

The  paper is well written and communicates the methods and outcomes of the study with clarity and brevity.

By and large the study was designed and analysed appropriately although was somewhat optimistic in the aims for detection of QTL given the limited progeny size and genetic markers used in the study.

Although I enjoyed the subject and clarity of presentation, I had concerns about interpretation of findings. The resistance phenotyping and characterisation of the parents and progeny is interesting and will be an excellent foundation for further study but when I got to the marker genotyping section I had concerns.

I was concerned about the large number of markers that had to be discarded due to genotyping “error”, and thus classified as non-transferrable, and the high degree of apparent segregation distortion of those that were used for mapping. The first issue of low transferability may be a consequence of poor quality DNA due to impurities, leading to failure of markers to amplify at all and thus they were discarded. Intermittent amplification of marker loci may have led to mis-classification of mating configurations and consequently the segregation distortion evident across almost all markers used for mapping. According to the genotype frequencies indicated in Fig 4, I cannot see how most markers were declared to fit the expected segregation ratio of 1:2:1 by Chi sq testing as claimed at  Line 139 and 184.

I expand on these issues below

Marker transferability

My main concern was the discarding the large number (39%) of markers due to “genotyping error” (Line 137). Later this is explained as markers with non-parental alleles in the progeny. Discussion (Line 274) addresses this but ropes it home to an issue with genotyping (allelic dropout; mis-priming, incorrect calling etc) although mentioning another possibility, i.e. pollen contamination. To me, unless there were major issues with the genotyping methodology (not possible to assess without seeing the PAGE gels), this large number of loci with non-parental alleles seems much more likely to be due to stray pollen (or mix up of seed during collection).

Given the closeness of the taxonomic affinity of pellita to the taxa where the markers were developed, (same subsection as urophylla) and same section as E. grandis, I think the expectation is for a fairly high rate of locus transfer, For example Brondani et al 2002 report 70% transportability of SSR loci within the subgenus Symphyomyrtus.

However this does not seem to have been the case in this study examining transfer to pellita.

The authors conclude markers which show non-parental alleles in progeny are due to non-transferrable markers but this doesn’t seem like the most probable explanation. If a progeny has a non-parental allele it is more likely that “progeny” is not a full sib, and could be a half sib in the case of stray pollen, or unrelated to either parent in the case of a “jumping” seed. If the loci is amplifying in some individuals in the pedigree, this suggests marker has transferred. Genotyping “error” does not seem the most appropriate term to describe the possible failure of the marker to conform with Mendelian inheritance patterns in this case.

Potential issues of pollen contamination or jumping seed

I believe you need to examine the patterns of individuals with non-parental alleles. Is there a particular subset of progeny which consistently show non-parental alleles? This would tend to support a conclusion that they aren’t full sib progeny, and are a consequence of pollen contamination and thus half sibs, or don’t belong to the cross at all (seed that.

DNA purity

DNA quality is another factor that may influence the reliability of genotyping. It is not possible to judge the quality of the DNA from the information provided but it may have been a a factor in apparent difficulty in marker transfer also.

Segregation distortion

Figure 4.You go to a lot of trouble to describe segregation distortion effects but don’t pick it up in the discussion. Is there not the possibility that your QTL coincide with regions of distortion on your map? There seems to be a correspondence in peak in your -log10- p-value with QTL locations but this depends on the orientation of the LG which is not possible to determine from your Figure without marker positions indicated on it. From your Fig 4 most markers would appear not to conform to expected marker segregation as the AA genotype frequency is close to Zero (if not zero) in most cases, where as it should be 0.25 right? Appears to be strong segregation distortion at almost all loci as AA genotype is rarely present. Suggests problems in genotyping and classifying markers into mating configurations. These markers can not fit a Chi square test for 1:2:1 if one homozygous class is zero. There should be about 25 individuals in a progeny set of 100 with the AA genotype.

Other comments

·         Line 140 Why are only markers in 1:2:1 ratio used for mapping? There are other configurations possible in this cross between two highly heterozygous parents ie. multiple intercross (MIC) with 4 and 3 alleles segregating, as well as backcross configuration 1:1. Were these configurations not seen or were they not considered? You may have been overlooking otherwise useful loci?

·         A loci with aMIC4 MIC3 or BC configuration could have also been evident? The MIC4 and 3 would have been informative. Table 2 – If only 1:2:1 markers (or those in an intercross configuration) are used for mapping, there should only be two alleles. Many of the markers in Table 2 are reported as having 3 alleles? Three alleles should give 4 progeny genotype each in equal frequencies, right?

·         Line 182 “Segregation distortion was within range expected”. This should be rephrased I think to something like, segregation ratios for markers were within X confidence level, where X is the value of the confidence level, often 95%. A large positive -log10 p-value is significant right? What is the threshold value for determining significance?

·         Calculation, reporting and Discussion of Ho and He in context of a single pedigree doesn’t make sense. Discard para beginning on Line 322 and remove from Table.

·         Line 114 DNA extraction and quality evaluation. How was the purity of the DNA evaluated? A spectrophotometric reading of A260/280 can be helpful in diagnosing impurity issues. A good test for purity for use in PCR is to perform a restriction enzyme digestion. If your DNA restricts it is almost certainly good for PCR. Poor quality DNA would cause non-amplification or mis-typing of genotypes but would be random wrt to QTL alleles, therefore shouldn’t cause segregation distortion.

·         Line 170 Need more clarity around threshold for usage of a marker needed. How many progeny amplified at each marker? I assume only markers which amplified in both parents were considered. Was a marker discarded if it did not amplify in one individual for example? Although good to have this strict threshold for accepting markers, need more context on the quality of DNA preparations. Discarding a marker that did not amplify from one progeny would seem too severe a cut off and would not allow for possibility of one poor quality DNA extraction.

·         Line 332 – This number matched…..   Which number matched? What expectation? Not at all clear.

·         Line 240. What were the PVE for each qtl? Ie phenotypic variance explained? This tells us something about the effect size, but which may be too reliable in this case anyhow due to the small population size.

·         Line 294 – Rationale for preference for dinucleotide repeats – This is generally attributed to their higher variability (more alleles) not that they “better explain the level of polymorphism”. They simply exhibit more polymorphism.

·         Line 288 – Microsatellite transferability – possibly higher than concluded here if those with non-parental alleles were thrown into the non-transferability pile?

·         Consider referring to QTL as “putative” QTL until verified in another cross. Couched in much more tentative terms at this early stage of discover.

Reviewer 2 Report

Summary

The authors present a very nicely written report on the development of a linkage map and QTL analysis of Ceratocystis resistance in E. pellita.  The motivation for such an analysis is the lack of genomic resources for this species of Eucalyptus.  Thus, SSR markers from other members of the genus were used (transferred) to this species for the construction of the linkage map and QTL analysis.  The authors also confirm that the approach to the development of a population for QTL analysis using parents from different locations and controlled pollination of full sibs is adequate to provide sufficient variation in resistance for analysis.   Overall the paper was easy to read, grammatically well done, and organized in a way that made understanding easy.

Major Criticism

I have no major concerns for this manuscript. I think some improvements to clarity are needed and perhaps some trimming of a few things.

Minor Criticism

On page 2, line 49 it says “Heritability estimates of more than 80% indicate that the best clones for a specific trait can be selected efficiently from eucalypt families.”  I’m a bit confused by this sentence. Are the authors suggesting that disease resistance to Ceratocystis has a high heritability and that’s what makes this trait appropriate for this type of analysis?  If so, can they reference the report that indicates heritability estimates for this trait?  This should perhaps be clarified.

Perhaps the authors should reword the hypotheses of lines 77-80 as they might not be specific.  To expound, it is not clear that the authors fully provide evidence to support these the way they are written.  For the first one, only two parents were used so they cannot make a claim about the wide range of resistance in the E. pellita progeny as a whole. For the second hypothesis the paper doesn’t test the hypothesis fully, more progeny from different parents would be needed to repeat the experiment. I think hypothesis #2 is really a working hypothesis that is okay to move forward on and is not something being tested.  For the 3rd hypothesis the authors do not do any validation, (e.g., through marker-assisted selection) so this is hypothesis is never actually tested.  I am probably being pedantic here, but for me, hypotheses must be specific and it must be clear that they are tested and that results show evidence to support them. My suggestion would be to remove these, they are not needed.

Some clarity is needed on line 87 when it reads “… the same system” . What system specifically is being referred to. Can the authors edit this to be more clear?

On line 88 some accession numbers are used but the readers are not told where these numbers are defined or housed.  If a reader wants to duplicate this experiment, then how would they get access to these lines?

On line 145 it reads “The same software was used for QTL mapping…”. What software specifically? Can the users edit this to be specific.  The only thing mentioned previously was that the analysis was performed in R. Specific packages, software and versions of these should be stated. 

On line 157 it says “mean Dis of the parents were…” I’m confused by this statement.  There were only two parents. A mean of 2 seems odd to me. Perhaps just state the exact DIs for each parent. 

Figure 2, the statement describing it on line 163, and the statement referring to that figure in the discussion on line 261 are misleading.  The paper makes it seem as though some new fact has been discovered, but the formula for calculating DI includes the lesion length. So, unless plant height varies widely there should be a correlation between DI and lesion length.  If plant height did vary widely then the DI formula most likely would not have been developed.  The statement that the “strong correlation… indicates either variable may be used for evaluating disease severity” is not news.   Figure 2 and the statements of correlation should be removed in my opinion.

The notation AA, AB and BB for allele analysis confuse me because some of the markers in Table 2 state they specifically have 3 alleles.   Was the analysis for Figure 4 only performed on the QTL markers?  Were those markers biallelic?

On line 240 it states that LOD of 13.9 is “significant”. The authors should state the threshold for determining the significance in the LOD score. 

Figure 5 and the interpretation confuse me.  On line 245 it says that in LG2 allele 2 is a maker for greater resistance, but the plot seems to show reduced resistance in the homozygous state. 

I think I understand that 17 of the 30 markers were used in 2 linkage groups, but I am not sure I understand that correctly and I think the authors should clarify in the manuscript.  There should be 11 linkage groups. Were only 2 shown because only 2 had QTLs so the other are ignored for convience?  Are only the 17 markers shown because these are in the 2 LGs that had QTLs so the authors are trying to focus only on those?  Stating clearly why the other 9 LGs are not show I think would help clarify for the reader.